# An Analysis of Genetic Polymorphisms in 76 Genes Related to the Development of Ovarian Tumors of Different Aggressiveness

**DOI:** 10.3390/ijms252010876

**Published:** 2024-10-10

**Authors:** Laura A. Szafron, Piotr Sobiczewski, Agnieszka Dansonka-Mieszkowska, Jolanta Kupryjanczyk, Lukasz M. Szafron

**Affiliations:** 1Maria Sklodowska-Curie National Research Institute of Oncology, 02-781 Warsaw, Poland; laura.szafron@gmail.com; 2Department of Gynecological Oncology, Maria Sklodowska-Curie National Research Institute of Oncology, 02-781 Warsaw, Poland; 3Cancer Molecular and Genetic Diagnostics Department, Maria Sklodowska-Curie National Research Institute of Oncology, 02-781 Warsaw, Poland; 4Department of Cancer Pathomorphology, Maria Sklodowska-Curie National Research Institute of Oncology, 02-781 Warsaw, Poland

**Keywords:** ovarian cancer, borderline ovarian tumor, DNA sequence variant, NGS, Western blot, *TP53*, *RAS*, *BRAF*, *BRCA1/2*

## Abstract

Borderline ovarian tumors (BOTS) are rare neoplasms of intermediate aggressiveness between cystadenomas and low-grade ovarian cancers (lgOvCa), which they share some molecular resemblances with. In contrast to the most frequent and well-described high-grade ovarian carcinomas (hgOvCa), the molecular background of BOTS and lgOvCa is less thoroughly characterized. Here, we aimed to analyze genetic variants in crucial tumor suppressors and oncogenes in BOTS (with or without the *BRAF* V600E mutation), lgOvCa, and hgOvCa in two gene panels using next-generation sequencing. Then, we verified the existence of selected polymorphisms by Sanger sequencing. Finally, Western blot analyses were carried out to check the impact of the selected polymorphisms on the expression of the corresponding proteins. Our study contributes to the molecular characterization of ovarian neoplasms, demonstrating divergent polymorphic patterns pointing to distinct signaling pathways engaged in their development. Certain mutations seem to play an important role in BOTS without the *BRAF* V600E variant (*KRAS*) and in lgOvCa (*KRAS* and *NRAS*), but not in hgOvCa. Additionally, based on multivariable regression analyses, potential biomarkers in BOTS (*PARP1*) and hgOvCa (*FANCI*, *BRCA2*, *TSC2*, *FANCF*) were identified. Noteworthy, for some of the analyzed genes, such as *FANCI*, *FANCD2*, and *FANCI*, *FANCF*, *TSC2*, the status of BRCA1/2 and TP53, respectively, turned out to be crucial. Our results shed new light on the similarities and differences in the polymorphic patterns between ovarian tumors of diverse aggressiveness. Furthermore, the biomarkers identified herein are of potential use as predictors of the prognosis and/or response to therapy.

## 1. Introduction

Ovarian carcinoma (OvCa) is a common and complex malignant disease with a generally poor outcome and an exceptionally high mortality worldwide [1]. There are two main types of OvCa: high-grade (hgOvCa) and low-grade (lgOvCa) ovarian carcinomas. The former is the most common type, characterized by extreme genomic instability, chromosomal rearrangements, and frequently mutated genes, especially those encoding tumor suppressor proteins, such as *TP53*, *BRCA1*, and *BRCA2* [2]. By contrast, lgOvCa is a rare ovarian tumor characterized by a younger age at diagnosis, relative chemoresistance, and prolonged survival compared to its high-grade counterpart. Additionally, lgOvCa do not bear or rarely have mutations in *TP53* and *BRCA1/2* [3,4], and they (especially those of the serous subtype) share molecular resemblances with borderline ovarian tumors (BOTS) [5]. BOTS are regarded as neoplasms that are less aggressive than invasive carcinomas. They are also a rare entity (about 15% of epithelial ovarian neoplasms) and have relatively low malignant potential. In contrast to OvCa, BOTS predominantly occur in women at a reproductive age, are usually diagnosed at a lower FIGO stage, and have better survival rates. Despite these advantages, BOTS are difficult to diagnose preoperatively by imaging methods because there are no specific criteria to distinguish them from their malignant counterparts with high confidence [6]. Additionally, following the complete removal of the tumor, even 20% of BOTS may recur. Most BOTS recur as borderline tumors; however, in about 30% of patients with peritoneal implants, OvCa develops [5,7,8]. In contrast to hgOvCa, in BOTS, mutations in *TP53* and *BRCA1/2* are rare [9,10,11], whereas the most frequently mutated genes are *BRAF* and *KRAS* (especially in BOTS of the serous subtype). Mutations in these genes are sometimes also found in serous lgOvCa, while, in patients with hgOvCa, their occurrence is rare [12,13]. In addition to *KRAS*, *BRAF*, *BRCA1/2*, and *TP53*, a few studies have also investigated the frequency of *PIK3CA*, *EGFR*, *CTNNB1*, *RAD51C*, *PALB2*, *CHEK2*, and *PTEN* mutations in BOTS compared to invasive ovarian carcinomas [14,15]. Nevertheless, data on the polymorphic status of tumor suppressors and oncogenes in BOTS remain scarce. OvCa are far better genetically characterized than BOTS. Still, there are discrepancies as to the clinical significance of some molecular markers that need to be dispelled.

Therefore, in BOTS without the *BRAF* V600E variant (here referred to as BOT), in BOTS harboring this genetic variant (BOT.V600E), and in lgOvCa and hgOvCa, we aimed to characterize the polymorphic status of 76 genes using two next-generation sequencing (NGS) gene panels. The first one comprised oncogenes and tumor suppressor genes involved in the development of hereditary ovarian cancer (41 genes) plus *CRNDE*, *IRX5*, and *CEBPA*. The second panel contained hot spots in genes frequently mutated in sporadic human cancers (37 genes), most of which were missing in the first gene panel. Except for the thorough examination of gene polymorphisms and their association with various clinicopathological parameters with the use of uni- and multivariable regression models, our workflow involved the confirmation of selected gene variants by Sanger sequencing, and also the verification of whether there is a correlation between the presence of the given polymorphism and the expression alterations of the corresponding protein. Hence, this work may contribute to a better understanding of the ovarian tumor molecular landscape and lay grounds for the discovery of new biomarkers.

## 2. Results

### 2.1. Distribution of Genetic Polymorphisms in Different Tumor Groups

After the NGS and bioinformatic analyses, we obtained a list of genetic variants with a high or moderate impact on the corresponding protein’s structure and function. Genetic alterations with these impacts were either analyzed in combination or separately to determine whether the variants with distinct impacts exhibit concordant or discordant effects on the ovarian tumor outcome. Additionally, two questions need to be clarified. Firstly, in the 44-gene panel, one extra oncogene, the investigation of which was unintended, *KCNMB3* [16], was enriched. This was probably because its locus partially overlaps that of the *PIK3CA* gene, which was originally included in the panel and encoded by the opposite DNA strand. Similarly, in the hot-spot panel, one extra gene, *FBXW7-AS1*, was enriched, being an antisense transcript of the *FBXW7* gene [17], present in the hot-spot panel. Secondly, polymorphisms in the *CRNDE* gene (enriched in the 44-gene panel) were earlier described in another paper by our team [18], and therefore they will not be addressed in this article.

When using the 44-gene panel in the entire group of ovarian tumors, we discovered 85 unique, previously undescribed variants (71 new SNPs and 14 new non-SNPs). The list of all the detected variants is presented in a Appendix A. The cumulative frequency of all the detected genetic variants (SNPs and non-SNPs combined) in different groups of tumors is presented in Figure 1A,B (variants with a high or moderate (A) or only a high (B) impact) and Appendix A (variants with a moderate impact only). In Figure 1C–F and Appendix A, relevant box plots, depicting the overall frequency of SNP and non-SNP variants separately in different ovarian tumor groups, are shown. These box plots are additionally supplemented with detailed statistical tests. Furthermore, the mean counts of either SNP or non-SNP alterations per gene per tumor group are also presented in Appendix A–F. When considering all the variants together (Figure 1A), the most frequently altered genes (the fraction of altered samples in at least one group > 0.5) were *BRCA1*, *BRCA2*, *FANCA*, *SEM1*, and *TP53.* However, if only high-impact variants are considered (Figure 1B), the highest frequencies of genetic alterations (>0.3) were present in the *BRCA1* and *TP53* genes, and in the hgOvCa group only. By contrast, the number of SNPs with a high/moderate (Figure 1C) or only a moderate (Appendix A) impact on a protein structure/function was significantly higher in the BOT without the *BRAF* V600E mutation compared to all the remaining tumor groups. Moreover, the same analysis revealed that the BOT.V600E tumors were characterized by a significantly lower number of SNPs than hgOvCa. Noteworthy, in hgOvCa, the number of high-impact genetic variants (either SNPs or non-SNPs) was significantly elevated compared to all the other tumor groups, except for SNPs with a high impact in the lgOvCa vs. hgOvCa comparison (Figure 1D–F). Remarkably, the numbers of non-SNPs with only a moderate impact on a protein structure/function did not significantly differentiate any ovarian tumor groups (Appendix A), conceivably due to the low frequency of these variants (six such changes were found in only three genes, *ATRX*, *CHEK1*, and *PTEN* (Appendix A), exclusively in nine hgOvCa tumors (see Appendix A)).

For the hot-spot panel, we discovered 82 unique, not previously described genetic variants (75 new SNPs and 7 new non-SNPs). Their list can be found in Appendix A. The cumulative frequency of all the detected variants (SNPs and non-SNPs combined) in all the genes in every group of tumors is presented in Figure 2A,B (variants with a high or moderate (A) or only a high (B) impact) and Appendix A (variants with a moderate impact only). In Figure 2C–F and Appendix A, relevant box plots, depicting the overall frequency of SNP and non-SNP variants separately in different ovarian tumor groups, are shown. These box plots are additionally supplemented with detailed statistical tests. Furthermore, mean counts of either SNP or non-SNP alterations per gene per tumor group are also presented in Appendix A–L. When considering high and moderate variants together, and with SNPs and non-SNPs combined (Figure 2A), the most frequently altered genes (the fraction of altered samples in at least one group > 0.5) were *PTCH1* (altered in all tumor groups) and *TP53* (altered mainly in hgOvCa). Interestingly, genetic variants in the *BRCA1* gene were less frequently identified in the hot-spot panel than in the 44-gene panel. Nevertheless, if only the high-impact variants are considered, the mutational profiles of *BRCA1* and *TP53*, detected with both panels, were similar, revealing the high frequency of genetic alterations within these genes in OvCa, especially in hgOvCa (Figure 1B and Figure 2B). Notably, in the hot-spot panel, we also detected two variants in the *TP53* gene in one lgOvCa sample (chr17:g.7674921C>A, p.Glu204Ter and chr17:g.7676218C>A, p.Glu51Ter). Neither of these SNPs were found in the 44-gene panel, probably due to their low frequencies, equaling 11% and 14%, respectively. It needs to be mentioned here that, in our bioinformatic workflow, all sequence variants less frequent than 10% were filtered out to eliminate alterations resulting from DNA polymerase errors and those too rare to both elicit an evident clinical effect and be successfully validated by Sanger sequencing.

When only SNPs are taken into account, the results for the hot-spot panel importantly differ from those obtained for the 44-gene panel with respect to the frequency of non-high-impact SNPs in the BOT group. In the hot-spot panel, the number of such SNPs in BOT was significantly lower than in both OvCa groups (Figure 2C and Appendix A). By contrast, in the 44-gene panel, non-high-impact SNPs in BOT were much more abundant than in all the remaining tumor groups (Figure 1C and Appendix A). Yet, this divergence disappeared when either high-impact SNPs or all non-SNPs were considered (Figure 1D–F and Figure 2D–F), revealing the increased frequency of genetic alterations in hgOvCa compared to BOTS in both panels.

All SNP and non-SNP variants per gene per sample were summed and binarized (at least one variant present vs. no alteration). The subsequent statistical analysis of this dataset, shown in Table 1, revealed that *TP53* was the most differentiating gene between less aggressive tumors (BOT, BOT.V600E, and lgOvCa) and hgOvCa (in the latter, it was more frequently mutated), regardless of the gene panel and the variant impact. The only exception to this rule was found for high-impact alterations in the lgOvCa vs. hgOvCa comparison in the hot-spot panel, where no statistical significance was observed.

Two other genes worth mentioning here are *BRCA1* and *BRCA2*, since, in this study, their mutational profiles seemed to depend not only on the gene panel used but also on the impact that the genetic alterations had on the structure and function of the proteins encoded by these genes. In the 44-gene panel, both aforementioned genes turned out to be more frequently altered in BOT than in hgOvCa if moderate-impact variants were considered. This regularity also persisted if high-impact alterations in the *BRCA2* gene were included. By contrast, only high-impact *BRCA1* variants occurred much more frequently in hgOvCa than in all the other ovarian tumor groups (Table 1). In the hot-spot panel, the *BRCA2* gene was not included, while the number of polymorphisms in *BRCA1* was significantly higher in hgOvCa than in BOT, irrespective of whether only high-impact or all genetic variants were taken into account.

In this study, *KRAS* was the gene in which high- or moderate-impact variants most strongly differentiated BOT from all the other tumor groups, except lgOvCa. In two other genes, involved in ubiquitination, *FANCB* and *SEM1*, moderate-impact variants were identified significantly more frequently in BOT than in OvCa (*SEM1*) or hgOvCa (*FANCB*). Of note, the variants in these two genes did not differentiate BOT from BOT.V600E. Moreover, from among 76 different genes investigated in the two panels in the present study, *BRAF* was the only one that was more frequently mutated in the BOT.V600E tumors compared to all the other groups.

Genetic changes in the *KRAS* gene occurred frequently not only in BOT but also in lgOvCa compared to BOT.V600E and hgOvCa. Apart from *KRAS*, variants in two other genes, *ATM* and *NRAS*, predominated in lgOvCa. ATM was more frequently altered in lgOvCa than in the BOT.V600E group, yet this regularity was confined to the moderate-impact variants only. As for *NRAS*, moderate-impact alterations in this gene prevailed in lgOvCa in comparison with the three remaining tumor groups.

For the confirmation of polymorphisms in the selected genes, we used gradient PCR combined with Sanger sequencing. With this technique, we managed to successfully verify one previously identified variant in the *TP53* gene (chr17:g.7670658_7670659insA, p.Lys351Ter) [19] and seven new variants (SNPs and non-SNPs) with either a moderate or high impact on a protein’s structure/function. The verification results and the detailed description of each analyzed polymorphism are presented in Appendix A.

### 2.2. Regression Analyses

Using the 44-gene panel, we identified that the genetic variants in *PARP1* were of prognostic value and had a significant impact on BOTS patients’ RFS (Table 2 and Figure 3A–D). Notably, no genetic variants in the genes investigated in this study were identified as good predictors of the occurrence of microinvasions or implants within the tumor masses in BOTS. In hgOvCa, the only marker found to be predictive of response to chemotherapy were genetic variants in *BRCA2.* Polymorphisms in this gene positively affected both the CR and PS in patients with tumors without the TP53 protein accumulation, either treated with TP or irrespective of the chemotherapeutic regimen used (Table 2 and Figure 3I). The genetic variants in *BRCA2* revealed their favorable prognostic value as well by decreasing the risk of death in the whole group of patients, in the subgroup treated with TP, and in patients with tumors without TP53 accumulation. The *FANCF* gene was discovered here as another marker of a good prognosis in hgOvCa, as its polymorphisms diminished the risk of death in the TP-treated patients with tumors lacking the TP53 accumulation. By contrast, in the same group of patients, the *FANCI* gene was identified as a negative prognostic factor, elevating the risk of relapse (Table 2 and Figure 3E–H).

In the hot-spot panel, no genetic variants of prognostic or predictive importance were found for BOTS. For hgOvCa, we discovered a single adverse prognostic marker, *TSC2*. Genetic variants in this gene increased the risk of death in patients treated with the TP regimen, whose tumors exhibited the accumulation of the TP53 protein (Table 2).

Of note, the regression analysis was not performed for lgOvCa patients due to the small size of this cohort (*n* = 10), making multivariable statistical inference impossible. Nevertheless, it is worth emphasizing here that in the randomization (chi-squared and Fisher’s exact) tests, described in Section 2.1, we managed to obtain statistically significant results for comparisons involving lgOvCa, which proved that the statistical power of these tests was high enough despite the rarity of lgOvCa tumors in our experimental setup.

### 2.3. Assessment of Relationship between Selected Gene Polymorphisms and Expression of Corresponding Proteins

To evaluate, on the protein level, the effects of the genetic alterations found in this study, we analyzed the expression of several proteins encoded by genes with SNP and non-SNP variants. The Western blot (WB) results are presented in Figure 4 and Figure 5. We observed a lower or no signal on a membrane for non-SNP frameshift polymorphisms detected in the following: *NBN* (chr8:g.89971217_89971221del, p.Lys219AsnfsTer16; Figure 4A); *CHEK2* (chr22:g.28695869del, p.Thr367MetfsTer15; Figure 4C); and *TP53* (chr17:g.7674900dup, p.Thr211AsnfsTer5; chr17:g.7670686del, p.Arg342GlufsTer3; chr17:g.7674241del, p.Cys242AlafsTer5; chr17:g.7676078del, p.Pro98LeufsTer25; chr17:g.7676041_7676042insTTTC, p.Arg110GlufsTer40; Figure 4E). Additionally, we analyzed some samples with *TP53* missense mutations (with a moderate impact) (chr17:g.7675085C>T, p.Cys176Tyr; chr17:g.7673824C>G, p.Gly266Arg; chr17:g.7676040C>G, p.Arg110Pro; chr17:g.7673776G>A, p.Arg282Trp), for which we observed TP53 accumulation and a strong signal on a membrane (Figure 4E). Interestingly, for *CHEK1* with a STOP-gain variant (chr11:g.125625996G>A, p.Trp79Ter; Figure 4G), a higher percentage of altered reads resulted in increased CHEK1 expression.

Moreover, we found out that the expression of FANCI and its protein partner, FANCD2, was mutually correlated and likely dependent on the presence of genetic variants in the *BRCA1/2* genes. Tumor cases with the *FANCI* chr15:g.89285210C>T (p.Leu605Phe) variant did not show any specific pattern of FANCI expression (Figure 5A). However, the same samples had a similar pattern of FANCD2 expression (Figure 5B), regardless of whether they harbored variants in *FANCD2* (Figure 5G). Yet, the occurrence of FANCD2 expression seemed to depend on the presence of *BRCA1/2* genetic alterations (Figure 5G). In the absence of sequence variants in these two genes, no signal for altered FANCI, and concomitantly for FANCD2, was observed on membranes (compare Figure 5G and Figure 5A,B). Additionally, we tested whether the most frequent variant in *FANCD2* (chr3:g.10073349G>T; p.Gly901Val) affected the FANCD2 expression, which revealed no relationship (Figure 5E,H).

## 3. Discussion

The aim of this study was the analysis of genetic variants in crucial tumor suppressors and oncogenes in ovarian tumors of different aggressiveness. We not only evaluated the polymorphic status of these genes in large, thoroughly characterized cohorts of OvCa and BOTS, but we also found predictive and/or prognostic markers for both tumor groups and analyzed the functional role of selected polymorphisms regarding their influence on the expression of the corresponding proteins.

Unexpectedly, our NGS results, obtained for the 44-gene panel, showed that the number of SNPs with a high or moderate or only moderate impact on the structure and/or function of the corresponding proteins was higher in BOT compared to BOT.V600E, lgOvCa, or hgOvCa. Conversely, when analyzing only hot spots in selected genes, the frequency of SNP variants with these impacts was significantly lower in BOT than in both OvCa groups. This apparent discrepancy may be explained by the fact that the two panels investigated in this study contained different sets of genes. As proven in the present study, the list of genes from the 44-gene panel more frequently mutated in BOT compared to the other tumor groups (*FANCB*, *SEM1*, *FANCA*, *BRCA2*, *CHEK2*, *MUTYH*, *RAD50*) was much longer than analogically altered genes in the hot-spot panel (*KRAS* only). Additionally, the hot-spot panel was designed to investigate well-known genetic alterations. By contrast, in the 44-gene panel, an approximately 10 times bigger region of the genome was covered, enabling the detection of rare genetic variants, usually omitted in, e.g., diagnostic approaches. Nevertheless, when only polymorphisms with a high impact on a protein function and/or structure were considered, the number of genetic variants identified in both panels was the highest in hgOvCa, thus supporting the general knowledge about ovarian carcinomas [20,21].

The mutational status of *TP53* can be considered one of the best markers differentiating hgOvCa (frequent mutations in *TP53*) from BOTS (no or very rare mutations) [11,22,23] and lgOvCa (relatively rare mutations) [24]. In line with these reports, TP53 was one of the most frequently altered genes in the present study, mainly in hgOvCa. By contrast, no variants in *TP53* were found in our lgOvCa cases, besides two poorly covered high-impact SNPs in one lgOvCa specimen. Interestingly, these SNPs were detected only in the hot-spot panel, making use of a novel NGS hybridization capture technology (known as Primer Extension Target Enrichment, KAPA HyperPETE, Roche), offering much better sequencing coverage uniformity than the older hybridization-based capture approach (KAPA HyperCap, Roche), utilized in the 44-gene panel [25]. As for BOTS, the only two missense variants in *TP53* found in this study were observed in two BOT samples of a mucinous subtype. This outcome aligns with the current state of the knowledge too, given that Kang et al. reported TP53 mutations in 19.4% of mucinous BOTS, which was associated with a higher risk of recurrence [26]. Consistently, one of our two *TP53* mutation-bearing BOT patients had progression to OvCa. Noteworthy, herein, we also managed to confirm our NGS results for *TP53* on the protein level by observing both the lack of TP53 in samples with high-impact non-SNP variants and TP53 accumulation in tumors harboring *TP53* missense SNPs. These results are in line with our previous immunohistochemical evaluation of TP53 expression [27].

According to the literature, alongside genetic aberrations in *TP53*, mutations in *BRCA1/2* are also frequent in hgOvCa [24,28] and are rare in BOTS and lgOvCa [3,9,10,24]. Our results obtained with the 44-gene panel do not seem fully consistent with the literature, as we found variants in *BRCA1/2* genes in many non-high-grade ovarian tumors. However, it needs to be emphasized that, except for one SNP in a single BOT, these were only moderate-impact variants. These variants accounted for the significantly higher number of genetic alterations found in *BRCA1/BRCA2* in BOT compared to hgOvCa. Yet, when only high-impact variants were considered, *BRCA1* (but not *BRCA2*) was, as expected, more frequently altered in hgOvCa in comparison with all the remaining tumor groups. By contrast, in the hot-spot panel, which concentrated on well-established variants only and omitted most of the poorly investigated genetic alterations, no *BRCA1* polymorphisms with a high/moderate impact were found in BOT or lgOvCa, and only a single moderate-impact variant was present in one BOT.V600E sample. As a consequence, when only commonly analyzed hot spots in the *BRCA1* gene were taken into account, our statistical workflow corroborated the generally acknowledged predominance of sequence variants in this gene in hgOvCa compared to BOT. Still, according to a recent NGS study, carried out on big cohorts (containing 1333 OvCa and 152 BOTS patients), the prevalences of *BRCA1/2* mutations are similar in hgOvCa and BOTS (30.9% and 28.9%, respectively) [29]. Thus, this paper seems to corroborate our finding, made with the 44-gene panel, that lots of genetic alterations in *BRCA1* are detectable in BOTS if high-throughput sequencing techniques (not limited to known hot spots only) are applied. As for *BRCA2*, similarly to *BRCA1*, the moderate-impact variants of this gene prevailed in BOT compared to hgOvCa. Conversely, we revealed no differences in the frequencies of high-impact *BRCA2* polymorphisms between the investigated groups of ovarian tumors. Nevertheless, *BRCA2* emerged in this study as a promising, favorable predictive and prognostic marker in hgOvCa. The presence of sequence variants in *BRCA2* improved the patient OS, CR, and PS, especially in tumors without TP53 accumulation. Although this outcome may seem odd, given the tumor suppressor capabilities of this gene, a similar phenomenon was earlier observed in small-cell lung cancer [30], where the authors of the cited research reported a link between the occurrence of *BRCA2* mutations and the higher sensitivity of tumors to chemotherapy. In line with these findings, data obtained in vitro also provided strong evidence for the better response of BRCA-deficient tumors to platinum drugs, which was further confirmed by ex vivo studies, where *BRCA* mutation carriers exhibited better survival and longer disease-free intervals upon treatment with platinum drugs [31]. As BRCA1/BRCA2 proteins are responsible for the repair of double-strand DNA breaks (DSBs), the presence of pathogenic variants in *BRCA2* leads to the impaired activity of its protein product and thus increases the risk of a DSB in a tumor cell. If such a cell expresses functional TP53 (no TP53 accumulation is observed), apoptosis is induced [32], thus ameliorating the outcome of platinum-based treatment, as shown herein.

As for genetic alterations characteristic of less aggressive ovarian tumors, the genes with the highest number of polymorphisms in BOTS and lgOvCa compared to hgOvCa were *KRAS*, *BRAF*, and *NRAS*, which is in line with the scientific literature [13,33,34,35,36]. Given that BOTS with the *BRAF* V600E variant occurred in much younger patients than those lacking this mutation [18], here, both these groups of tumors were analyzed separately. Interestingly, *KRAS* was more frequently mutated in BOT and lgOvCa than in either BOT.V600E or hgOvCa, while the frequencies of *KRAS* variants in lgOvCa and BOT were comparable. This confirms the molecular resemblance between these two tumor groups. Simultaneously, such an outcome demonstrates that in BOTS without the *BRAF* V600E variant (being the most frequent polymorphism in this gene, found in this study in about 72% of *BRAF*-deficient tumors), KRAS-activating mutations are present. The KRAS-dependent cancer-promoting mechanism hinges mainly on mutations in the Gly12(G12)-coding region of the gene [37,38], which, in our research, predominated in BOT and lgOvCa alike. By contrast, none of the *KRAS* polymorphisms, which we found in a few hgOvCa tumors, affected Gly12. Furthermore, it is worth mentioning that all three of our BOT cases with *BRAF* variants other than V600E (i.e., K601E, G466R, and G466V) simultaneously harbored *KRAS* G12 variants. This suggests that out of all the *BRAF* polymorphisms, only *BRAF* V600E exerts a sufficiently strong cancer-promoting effect to act independently of *KRAS* mutations [39]. As for *NRAS* variants, their prevalence differentiated lgOvCa from all the other tumor groups investigated in our study. This outcome supports the finding of others that mutations in *NRAS* are found in serous lgOvCa but not, or rarely, in serous BOTS [40]. Similarly to activating mutations in *KRAS*, their counterparts in *NRAS* also speed up tumor progression. Moreover, such variants are found in recurrent serous lgOvCa too [41,42]. In this context, it is worth mentioning that one of our serous BOT samples with microinvasions harbored the NRAS-activating variant (p.Gln61Arg) [43], which occurred most frequently in our lgOvCa group as well. The presence of such a mutation in a BOT sample not only constitutes further confirmation of the molecular similarity between BOT and lgOvCa [5,44] but also implies that this BOT tumor might have transformed and recurred as lgOvCa if it had not been completely excised. According to the literature, in advanced ovarian carcinomas, *NRAS* mutations are rare [45]. Consistently, we did not identify such genetic alterations in our hgOvCa series. Of note, mutations in the *KRAS*, *NRAS*, and *BRAF* genes have also been reported in other human malignancies, e.g., colorectal and endometrial cancers [46,47,48,49].

Genes encoding proteins involved in ubiquitination were also more frequently altered in BOT and differentiated these tumors from OvCa (but not from BOT.V600E). One of these genes, *SEM1*, which codes for a 26S proteasome subunit [17], was very often altered in all the tumor groups. Although the most frequent variant, found in all the tumor groups, p.Gln59Pro is widespread in the human population (maximum allele frequency (AF_max_) of 0.88); still, the overall number of *SEM1* variants was significantly higher in BOT than in either lgOvCa or hgOvCa. Nowadays, no scientific reports on the role of this polymorphism in tumors are available. For the second gene, *FANCB*, which encodes a DNA repair-involved protein required for FANCD2 ubiquitination [17], literature data concerning OvCa are scarce, while its function in BOTS has not been studied so far. *FANCB* missense mutations were shown to cause the instability of the catalytic module and Fanconi Anemia (FA) core complex dysfunction. By contrast, SNPs in the *FANCB* 3′UTR did not affect the expression or function of the protein [50]. Given that all the *FANCB* polymorphisms found in our research were located in the coding sequence of the gene, their occurrence may likely impair the FANCB function, as proven in the study cited above. Interestingly, according to the current state of the knowledge, the *FANCB* role in cancer seems discrepant. On the one hand, no mutations in this gene in hereditary breast/ovarian cancers were found [51] and no associations between *FANCB* and the development of *BRCA1/2*-negative familial cancers were demonstrated [50]. On the other hand, Matta et al. [50] unraveled the relationship between the expression of *FANCB* and breast cancer in older patients with decreased DNA repair capacities. In this context, our results appear to shed new light on the clinical importance of *FANCB*, showing that this gene may play more important roles in BOTS than in OvCa.

Our regression analysis revealed genetic variants in *PARP1* as a marker of a poor prognosis in BOTS. This gene encodes a protein activated by DNA damage, regulating the function of many tumor suppressors, including TP53 [52]. In the literature, the data on the *PARP1* role in BOTS are limited; however, its meaning in OvCa has been profoundly investigated [53,54]. Consequently, PARP inhibitors have been approved for the maintenance treatment of recurrent platinum-sensitive *BRCA1*/*2*-deficient OvCa. Yet, newer data demonstrated therapeutic benefits in tumors beyond those with *BRCA1/2* mutations [55]. Remarkably, the most frequent *PAPR1* polymorphism in all the groups of tumors analyzed herein, p.Val762Ala, was different from that causing resistance to olaparib, one of the PARP inhibitors [56]. Despite its predominance in the human population (AF_max_ around 45%), the p.Val762Ala variant was previously shown to be associated with several types of cancer, including gallbladder cancer [57,58]. The same polymorphism also increased the risk of breast cancer among the Saudi and Asian populations, simultaneously decreasing this risk among Caucasians [59]. Interestingly, though other scientists reported that *PARP1* expression in serous OvCa is higher than in BOTS [60], in our hgOvCa series, this gene was neither more frequently altered nor identified as a potential biomarker.

Polymorphisms in two other genes encoding proteins involved in the FA pathway, FANCF and FANCI, were identified herein as promising outcome predictors in hgOvCa. Noteworthy, variants in *FANCI* exhibited significantly better discriminative capabilities than those in *FANCF*, as assessed based on the AUC values. The FANCI protein forms a heterodimer with FANCD2, which is subsequently monoubiquitinated by the FA core complex. Such a heterodimer localizes to the damaged chromatin and promotes interstrand crosslink repair [50]. In our analyses, the presence of variants in the *FANCI* gene increased the risk of recurrence in the TP-treated patients with tumors lacking the TP53 accumulation. When the literature data are considered, the role of *FANCI* seems ambiguous, as this gene has been reported to play both oncogenic and tumor suppressor roles [61,62]. Moreover, *FANCI* was recently proposed as a new OvCa-predisposing gene in carriers of the *FANCI* p.Leu605Phe variant [63], the frequency of which turned out to be significantly higher in OvCa-prone families with normal *BRCA1/2* genes [64]. In vitro studies revealed that the Leu605Phe isoform of FANCI was expressed at a reduced level and conferred sensitivity on HeLa and OvCa cells to cisplatin but not to a PARP inhibitor [64]. Consistently, our WB analyses revealed that tumors with the *FANCI* p.Leu605Phe variant and normal *BRCA1/2* genes did not express mutated FANCI, in contrast to *BRCA1/2*-deficient tumors, where FANCI expression was detected. Additionally, the same WB analysis unraveled the correlation between the expression of the FANCI and FANCD2 proteins. All these results clearly suggest that the role of *FANCI* depends on the molecular background in the cell controlled by crucial tumor suppressors, such as BRCA1/2 and TP53.

Our last result worth discussing deals with *CHEK1* for the nonsense variant in which (chr11:g.125625996G>A, p.Trp79Ter) we observed the unexpectedly high expression of the CHEK1 protein. Interestingly, both molecular phenomena seemed to be positively correlated (the higher the percentage of the altered allele, the stronger the signal for CHEK1 on a membrane). The SNP in question is located in the first exon/5’UTR region of *CHEK1*. If the longest isoform of CHEK1 (XP_011540862.1) is considered, the discussed polymorphism leads to the formation of a premature stop codon. In such a case, the utilization of an alternative start codon located downstream from the newly formed stop codon may not only restore the CHEK1 expression as a shorter isoform but concomitantly affect its levels in the cell. Consistently, according to the literature, short CHEK1 isoforms may occur due to alternative splicing or protein cleavage [65]. The role of *CHEK1* in tumorigenesis is ambiguous. Initially, CHEK1 was thought to be a tumor suppressor because of the role it plays in the DNA damage response and cell cycle checkpoint response [66]. However, no evidence of homozygous loss of function *CHEK1* mutants in human cancers was found. Moreover, the *CHEK1* gene was overexpressed in several solid tumors, and its expression was correlated with the tumor grade and disease recurrence [67]. In step with these findings, the complete loss of *CHEK1* suppresses chemically induced carcinogenesis, whereas tumor cells with increased levels of *CHEK1* may acquire survival advantages due to the ability to resist chemotherapy-induced DNA damage. As a result, reduced survival rates of patients with high *CHEK1* expression were reported in bladder, brain, lung, ovary, and breast cancers [67]. Although our results do not elucidate whether CHEK1 acts more like an oncogene or suppressor in ovarian tumors, further investigation of its variants appears interesting in the context of potential targeted therapies with Prexasertib, a selective CHEK1 inhibitor. Its application, either as a single agent or in combination with PARP inhibitors, stimulated tumor regression and prolonged hgOvCa patient survival [68]. This combination of inhibitors could be of potential use in BOTS, since *PARP1* polymorphisms were identified herein as a negative prognostic marker in these tumors, while some BOTS also harbored the above-described *CHEK1* p.Trp79Ter variant.

Finally, as with every study, this one also has some limitations that ought to be mentioned here. Although we managed to identify numerous genetic variants, due to financial and time-related constraints, the functional validation was only performed for a small subset of these polymorphisms. Thus, the clinical significance of many identified variants, listed in the Appendix A, remains unclear and should be addressed in future research. Furthermore, it needs to be emphasized that in our bioinformatic workflow, all sequence variants less frequent than 10% were filtered out. This approach was utilized to reduce the rate of false-positive hits, yet, hypothetically, some rare, clinically important polymorphisms may have been excluded from the analysis too. The next limitation worth bringing up results from the fact that we analyzed bulk tumor samples, which are just a part of the entire tumor microenvironment, the complexity and heterogeneity of which might not have been fully captured due to the constraints of the experimental setup applied in this study. Also, in terms of the tumor complexity and heterogeneity, we are aware that the loading controls in our Western blot experiments sometimes differed between lysates from distinct OvCa samples analyzed on the same gel. This inconsistency was not caused by any laboratory error or imprecision but, rather, is related to the vast biodiversity of ovarian tumors, especially high-grade OvCa, which results from the genomic and proteomic instability of such malignancies [69]. In the present study, to diminish the risk of drawing false conclusions, the concentration of all the protein lysates was not only assessed by Ponceau S red staining but was also precisely measured and normalized with the BCA method and a standard curve for bovine serum albumin (BSA). In the end, the present research was performed on a retrospective (not prospective) cohort of patients, collected for 20 years, meticulously followed up, and carefully checked for the compatibility of all the clinicopathological parameters. This approach, though widely used, could introduce some hardly definable biases and limit the ability to control for potential confounding factors.

## 4. Materials and Methods

### 4.1. Patients and Clinicopathological Parameters

In this study, a retrospective set of 225 non-consecutive ovarian tumor samples was used, including 76 BOTS (61 of the serous type and 15 of other histological types), 10 lgOvCa (9 of the serous type and 1 of another type), and 139 hgOvCa (113 of the serous type and 26 of other types). All the samples were collected from an ethnically uniform cohort of patients of central European origin, hospitalized at the Maria Sklodowska-Curie National Research Institute of Oncology, Warsaw, Poland, in the years 1995–2015. The corresponding medical records were critically reviewed by at least two physicians. Out of 76 BOTS, 21 were collected as snap-frozen samples, whereas the remaining 55 specimens were available in the form of formalin-fixed, paraffin-embedded (FFPE) blocks only. By contrast, all our OvCa samples were snap-frozen. The detailed clinicopathological characteristics for the BOTS and OvCa are presented in Appendix A, respectively. For two lgOvCa, the information on the applied chemotherapy was missing, which was one of the grouping variables in our study. Therefore, these samples were excluded from Appendix A. As for the evaluation of the clinical endpoints, all surviving patients had at least a 3-year follow-up. The specimens were carefully selected to meet the following criteria: an adequate staging procedure (stages were assessed for all cancers and primary BOTS) according to the recommendations by the International Federation of Gynecologists and Obstetricians (FIGO) [70], tumor tissue from the first laparotomy available, availability of clinical data including patient age and follow-up, as well as tumor histological type and grade and residual tumor size. Noteworthy, all BOT patients were characterized by no residual disease. All tumors were uniformly histopathologically reviewed and classified according to the new WHO criteria [5,71]. Additionally, a complete evaluation of the genetic variants in the *TP53* gene (for all tumors) and the TP53 protein status (for cancers only) was performed by either next-generation sequencing or with the PAb1801 mouse monoclonal antibody (1:500, Sigma-Genosys, Cambridge, UK), as described previously [27]. Most BOT patients (*n* = 60) did not undergo any chemical treatment. The remaining individuals suffering from BOTS (*n* = 16) received chemotherapy, administered either pre- or postoperatively. All carcinomas were excised from previously untreated patients. A total of 35 OvCa patients were treated postoperatively with platinum/cyclophosphamide (PC), while 112 of them underwent the taxane/platinum (TP) treatment after a surgical intervention. In BOTS, the relapse-free survival time (RFS) and the presence of microinvasions or implants within the tumor masses were used as dependent variables determining the disease outcome. The chemotherapy administration status was used as an independent logical variable in the multivariable statistical analyses. Other covariates taken into account in the multivariable statistical inference in BOTS were a logical variable determining whether the tumor was primary, the tumor histological type, and the patient age (continuous variable). In addition, BOTS were analyzed in the entire cohort of patients, and in subgroups comprising either BOT.V600E or BOT specimens only, since the presence of the *BRAF* V600E mutation was previously found to be significantly correlated with the lower age of patients diagnosed with BOTS [18]. For cancers, the overall survival (OS) and disease-free survival (DFS) of patients were used as dependent prognostic variables, while the platinum sensitivity (PS) and complete remission (CR) served as dependent factor variables predictive of the tumors’ response to treatment. CR was defined as the disappearance of all clinical and biochemical symptoms of ovarian cancer assessed after completion of the first-line chemotherapy and confirmed four weeks later [72]. DFS was assessed only for the patients who achieved CR. As for the independent variables used in the statistical analyses in cancers, the histological type and clinical stage of the tumors along with the residual tumor size were taken into account as factor variables in the multivariable statistical models. Noteworthy, due to the small size of the lgOvCa subgroup, only hgOvCa samples were subjected to the regression analyses performed in the present study. The hgOvCa were investigated in either the entire set of samples or in subgroups depending on the chemotherapy regimen used (PC/TP) and/or the TP53 accumulation status. Notably, two of the above-mentioned lgOvCa samples excluded from Appendix A were taken into account in the entire bioinformatic workflow presented herein, except for the Cox and logistic regression analyses, which required detailed clinicopathological information.

### 4.2. DNA Isolation and Quality Assessment

Genomic DNA (gDNA) from snap-frozen sections was isolated using the QIAmp DNA Mini Kit (Qiagen; Hilden, Germany), whereas gDNA from FFPE blocks was extracted on the MagCore Nucleic Acid Extractor machine using the MagCore Genomic DNA FFPE One-Step Kit (RBC Biosciences, Xinbei City, Taiwan). gDNA concentrations were measured on the Qubit 4 Fluorometer (Thermo Fisher Scientific (Thermo), Waltham, MA, USA) using the Qubit dsDNA HS Assay Kit (Thermo). Before the construction of the NGS libraries, the gDNA quality was assessed using our in-house-developed method based on the comparison of the real-time quantitative PCR (qPCR) efficiency for two amplicons of different lengths, described in the paper by Woroniecka et al. [73].

### 4.3. Construction of Total gDNA Libraries; 44-Gene Panel Enrichment and Verification; NGS Sequencing

For the libraries’ construction, 120–500 ng of gDNA was used. Libraries were created using the KAPA Hyperplus Kit (Roche, Basel, Switzerland) according to the protocol provided by the producer. The verification of the libraries’ size was made on 2100 Bioanalyzer (Agilent Technologies, Santa Clara, CA, USA). Total gDNA libraries were then enriched in exonic sequences of the following 44 genes: *ATM*, *ATR*, *ATRX*, *BAP1*, *BARD1*, *BCL2L1*, *BLM*, *BRCA1*, *BRCA2*, *BRIP1*, *CCNE1*, *CEBPA*, *CHEK1*, *CHEK2*, *CRNDE*, *EMSY*, *FANCA*, *FANCB*, *FANCC*, *FANCD2*, *FANCE*, *FANCF*, *FANCG*, *FANCI*, *FANCL*, *FANCM*, *IRX5*, *MDM2*, *MRE11*, *MUTYH*, *NBN*, *PALB2*, *PARP1*, *PIK3CA*, *PRKDC*, *PTEN*, *RAD50*, *RAD51B*, *RAD51C*, *RAD51D*, *RAD54L*, *RPA1*, *SEM1*, and *TP53*, using the SeqCap EZ Hybridization&Wash Kit with biotinylated hybridization probes (Roche). Out of these genes, 41 were involved in hereditary ovarian carcinoma development (as stated in the description of the Ion AmpliSeq™ Comprehensive Ovarian Cancer Research Panel, Thermo). The remaining three genes, *CRNDE*, *IRX5*, and *CEBPA*, were added by our team to further extend the functionality of this panel. The whole enriched region covered ca 360,000 bp in the genome. The verification of the DNA enrichment was performed by qPCR with four pairs of primers designed by Roche. The list of primers and the results of the enrichment evaluation for each primer pair are presented in Appendix A–D,F. The NGS libraries were sequenced on the NovaSeq 6000 Platform (Illumina, San Diego, CA, USA) in the paired-end mode (2 × 100 bp for DNA obtained from frozen material or 2 × 75 bp for DNA isolated from FFPE blocks). The resultant BAM files were deposited in the European Nucleotide Archive (ENA) database (data acc. no. PRJEB75542).

### 4.4. Hot-Spot Panel Enrichment and Verification; NGS Sequencing

For the hot-spot analysis, total gDNA libraries, also employed for the 44-gene panel, were used. The enrichment in 37 genes frequently mutated in sporadic human cancers (*AKT1*, *ALK*, *APC*, *ATM*, *BRAF*, *BRCA1*, *CDKN2A*, *CTNNB1*, *EGFR*, *ERBB2*, *ESR1*, *FBXW7*, *FGFR1*, *FGFR2*, *FGFR3*, *GNA11*, *GNAQ*, *GNAS*, *HRAS*, *IDH1*, *IDH2*, *JAK2*, *KIT*, *KRAS*, *NF1*, *NRAS*, *NTRK3*, *PDGFRA*, *PIK3CA*, *POLE*, *PTCH1*, *PTEN*, *RET*, *STK11*, *TP53*, *TSC1*, *TSC2*) was performed using the KAPA HyperPETE Hot Spot Panel (Roche). The whole enriched region covered approximately 36,000 bp in the human genome. The verification of the gDNA enrichment was performed using qPCR with one pair of our in-house-designed primers for *TP53* exon 4. For the enrichment verification results and PCR primer sequences, refer to Appendix A. The NGS libraries were sequenced on the iSeq100 platform (Illumina) in the paired-end mode (2 × 150 bp for DNA obtained from frozen material or 2 × 100 bp for DNA isolated from FFPE blocks). The resultant BAM files were deposited in the ENA database (data acc. no. PRJEB75531).

### 4.5. Bioinformatic Analyses

The quality of our NGS data (FASTQ files) was assessed with the FASTQC app (v. 0.12.1) and then optimized with Trimmomatic (v. 0.39). Mapping to the reference human genome (hg38) was performed using the HISAT2 aligner (v. 2.2.1). Afterward, the mapping quality was evaluated with the Samtools (v. 1.6), Genome Analysis Toolkit (v. v4.5.0.0), and Qualimap (v. 2.3) apps. Next, our in-house-developed software, SeqDepth_checker (v. 1.0, downloadable from https://github.com/lukszafron, LMS_gh, accessed on 29 May 2024), was utilized to evaluate the mean sequencing read coverage depths for each region enriched in every gene. If the mean coverage depth for a given region was lower than 5, this region was excluded from further analyses to diminish the risk of considering unevenly enriched DNA regions as non-mutated in samples with poor enrichment. The obtained BAM files were subsequently analyzed with bcftools software (v. 1.18) to create VCF files with the AD tag. Next, the variants were subjected to two-step filtering. First, variants less frequent than 10% were filtered out based on the AD tag, using the VAF checker app (version: 1.0), a program available for download at LMS_gh. Then, the vcf-annotate app from the VCFtools package (version: 0.1.16) was employed to filter out variants that did not meet the following criteria: all filters with default values applied, except for MinAB = 2 (a minimum number of alternate bases of 2), Qual = 20 (minimum sequence quality of 20), MinMQ = 20 (minimum mapping quality of 20), and MinDP = 5 (minimum sequence coverage depth of 5). Subsequently, the obtained VCF files were divided with bcftools into two subsets, SNPs and non-SNPs, containing SNP variants vs. all other sequence alterations, i.e., indels (insertions, deletions), mnps (multi-nucleotide polymorphisms), bnd (breakpoints), and others, respectively. Next, the variant identification and effect prediction analysis was carried out using the Ensembl Variant Effect Predictor (VEP) app (v. 109) and the merged Ensembl and RefSeq databases [74]. The obtained tab-delimited CSV files (VEP output tables) were further analyzed consecutively with two R programs developed by LMS, vep.r (v. 2.2) and vep.comparison.r (v. 2.2), both available for download at LMS_gh. Ensembl VEP divides sequence variants into four categories: high, moderate, low, and modifier, based on their expected impact on the transcript and protein sequences. For details, refer to the Ensembl web page [75]. The two aforementioned R apps were utilized first to filter out all variants characterized by low or modifier impacts and then to exclude all variants except those that either had a known adverse clinical significance (determined with the CLIN_SIG tag) or negatively affected the protein structure and function (as assessed by either the SIFT or PolyPhen algorithms). The new, previously unidentified sequence variants (with an empty “Existing_variation” field in the VEP output table), variants for which all three “CLIN_SIG”, “SIFT” and “PolyPhen” fields were empty, or those with a maximum allele frequency (MAX_AF) lower than 0.01, were also included in the final report generated by the vep.r app. The analyses were carried out independently for SNP and non-SNP variants. Subsequently, these results were combined with the binarization of sequence alterations for every gene (sequence variants with a high or moderate impact present (1) vs. absent (0)). Afterward, to identify genes with significantly different frequencies of sequence alterations between the investigated groups of ovarian tumors, statistical inference with the chi-squared test or the Fisher’s exact test (depending on the sizes of the analyzed subgroups) was carried out, followed by the data visualization. This final step of the analysis was performed with the vep.comparison.r script. A list of all polymorphic variants for each sample is presented in Appendix A.

All genes containing variants identified in our bioinformatic analyses were subsequently subjected to detailed statistical inference with the use of univariable and multivariable Cox proportional hazards models (package: survival, v. 3.5.7) to assess the value of these genes as potential novel prognostic biomarkers. All Cox models were also checked with respect to the proportionality of hazards for each variable used. The prediction of the treatment response was carried out by generating univariable and multivariable logistic regression models (packages: stats., v. 4.0.2, and rms, v. 6.0.1). The dependent, independent, and grouping variables (different for BOTS and hgOvCa) are described above in the section entitled Patients and Clinicopathological Parameters. In order to verify the discriminating capabilities of the created Cox and logistic regression models, we performed their cross-validation in new datasets, generated from the original data by bootstrapping (with replacement) and a subsequent comparison of the areas under ROC curves (AUCs) between the original and bootstrapped datasets, using the riskRegression package (v. 2023.12.21) [76]. The R script written to automate the above-mentioned statistical inference and subsequent visualization of the results (regression.analyses.r, v. 1.2) is downloadable from LMS_gh.

In order to identify the best potential biomarkers, we performed a matching of our regression analyses’ results. In this step, each univariable model was compared with its multivariable counterpart, and the models were considered matched when the analyzed genes and groups of tumors were the same, when both *p*-values were <0.05, when both HR/OR values were either higher or lower than 1, and, concomitantly, when the discriminating capabilities of both models were good enough (AUC values >0.65). Notably, in this paper, only the models that matched are presented.

### 4.6. Verification of Selected Polymorphisms

In this study, the following selected genetic variants (with coordinates consistent with the hg38 human genome assembly) in 8 genes were verified by gradient PCR and Sanger sequencing: *MUTYH*, chr1:45332673del; *BRCA2*, chr17:43093093_43093096del; *FANCE*, chr6:g.35456000T>G; *FANCI*, chr15:g.89295051C>T; *FANCM*, chr14:g.45187852C>G; *PRKDC*, chr8:g.47779009C>T; *RAD51D*, chr17:g.35106436del; and *TP53*, chr17:g.7670658_7670659insA, COSV99037094. The PCR reactions employed either the AmpliTaq Gold™ DNA Polymerase (Thermo) or the Phusion Green High-Fidelity DNA Polymerase (Thermo)) and in-house-designed sets of primers (Appendix A). PCR products were analyzed by agarose gel electrophoresis using the Simply Safe reagent (EurX, Gdansk, Poland) for DNA visualization. Gels were documented on the UVP ChemStudio Imaging System (Analityk Jena, Jena, Germany). Afterward, specific PCR products of expected lengths were cleaned with ExoSAP-IT (Thermo) and sequenced using the appropriate primer and the BigDye Terminator v 3.1 Cycle Sequencing Kit (Thermo). Sanger sequencing products were then cleaned with the ExTerminator Kit (A&A Biotechnology, Gdansk, Poland) and analyzed on the 3500 Genetic Analyzer (Thermo).

### 4.7. Protein Concentration Measurement

Total protein lysates were obtained by incubating tumor samples with the RIPA buffer supplemented with the Halt Protease Inhibitor Cocktail (Thermo). Next, the concentration of each lysate was evaluated with the BCA assay (Sigma Aldrich, Saint Louis, MO, USA), using BSA (Thermo) in amounts ranging from 0 to 25 µg per well as a standard curve. The absorbance at 540 nm was measured on the Victor 3 spectrophotometer (model: 1420-012, Perkin Elmer, Waltham, MA, USA). The negative control wells, containing only the BCA solution, were used as blank samples in this experiment.

### 4.8. Western Blot (WB) Analyses

WB analyses were performed for selected variants in genes coding for the TP53, NBN, CHEK1, CHEK2, FANCI, and FANCD2 proteins. Each WB experiment was preceded by WB tests confirming the specificity of the used primary antibodies (Abs). Except for lysates prepared from tumors, we also used a lysate prepared from a normal ovary as a control. For each experiment, 15–20 µg of a protein lysate was added per well. An electrophoretic separation of proteins was performed in 10–12% polyacrylamide gels (40% stock solution with the acrylamide to bis-acrylamide ratio equaling 37.5:1, BioRad, Hercules, CA, USA). To estimate the molecular weights of proteins, we used either the Broad Range Prestained Protein Marker (Proteintech, Rosemont, IL, USA) or the Precision Plus Protein Standard (BioRad). Depending on the protein being analyzed, either 0.2 μm nitrocellulose (Amersham™ Protran^®^, Cytiva, Marlborough, MA, USA) or 0.2 μm PVDF (Thermo) membranes were used. The transfer buffer was composed of 25 mM Tris (Sigma Aldrich), 192 mM Glycine (Sigma Aldrich), and 5–10% (*v*/*v*) methanol (Sigma Aldrich). Protein transfer was performed overnight (4 °C, 27 mA) or for 1–1.25 h (4 °C, 300 mA). For the membrane blocking, a 5% solution of skimmed milk (SM Gostyn, Gostyn, Poland) in the 1xTBST buffer (Tris-buffered saline (0.05 M Tris and 0.15 M NaCl) with 0.1% Tween-20 detergent (Sigma)) was used. As loading controls, Ponceau S red (Sigma Aldrich) staining, rabbit anti-β-actin Ab (1:100) (Thermo), and rabbit anti-vinculin Ab (1:500) (Thermo) were applied. Most primary Abs against selected proteins were purchased from Proteintech and were polyclonal antibodies developed in rabbits. By contrast, the primary mouse anti-TP53 antibody (Calbiochem, San Diego, CA, USA) was monoclonal. Chemiluminescence signals were detected on the UVP ChemStudio Imaging System (Analytik Jena, Jena, Germany) using either the goat anti-rabbit HRP-conjugated secondary Ab (Thermo) or the goat anti-mouse HRP-conjugated secondary Ab (Proteintech) and the SignalBright Max Chemiluminescent Substrate (Proteintech). A detailed description of the WB conditions for each protein is presented in Appendix A.

## 5. Conclusions

In this study, we examined the role of polymorphic variants in the most important oncogenes and suppressors in BOTS, lgOvCa, and hgOvCa. Our work contributes to the elucidation of the molecular landscape of various ovarian neoplasms, demonstrating completely divergent mutation profiles and molecular pathways engaged in their development. Certain mutations seem to play an important role in BOTS without the *BRAF* V600E variant (*KRAS*) and in lgOvCa (*KRAS* and *NRAS*), but not in hgOvCa, once again proving that advanced OvCa are molecularly distinct from less aggressive ovarian neoplasms. Additionally, based on multivariable regression analyses utilizing detailed clinicopathological data, potential biomarkers in BOTS (*PARP1*) and hgOvCa (*FANCI*, *BRCA2*, *TSC2*, *FANCF*) were identified. Noteworthy, for some of the analyzed genes, such as *FANCI*, *FANCD2*, and *FANCI*, *FANCF*, *TSC2*, the status of BRCA1/2 and TP53, respectively, turned out to be crucial. Although thorough mechanistic insight is necessary to fully investigate the molecular background of each genetic variant reported herein and to understand its clinical importance, still, our work sheds new light on the similarities and differences in the polymorphic patterns between ovarian tumors of diverse aggressiveness. Thus, it forms a valuable foundation for future research.

## Figures and Tables

**Figure 1 ijms-25-10876-f001:**
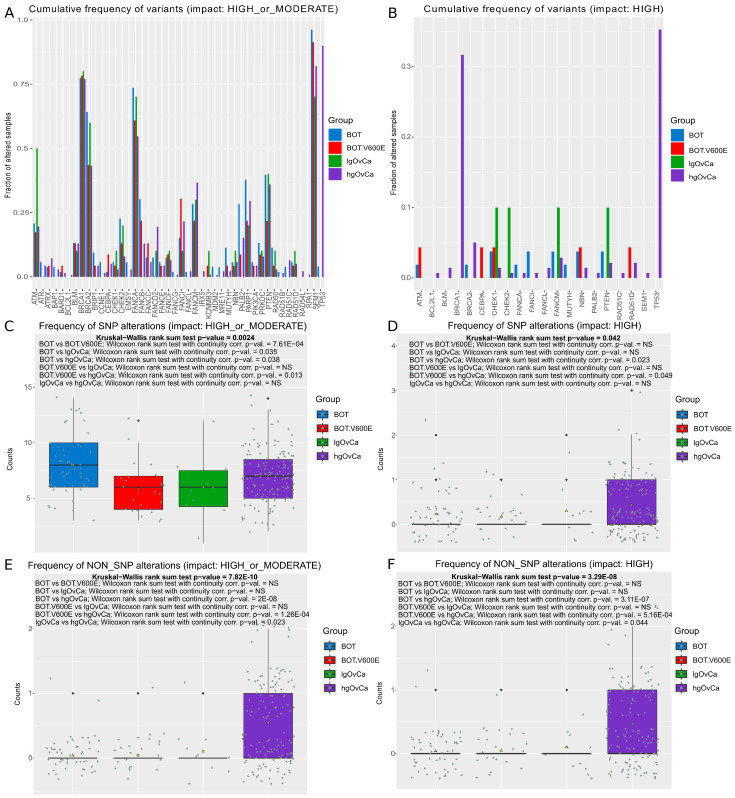
SNP and non-SNP variants—44-gene panel. (**A**,**B**) The cumulative frequency of all variants per gene in each tumor group ((**A**)—variants with a high or moderate impact, (**B**)—only the variants with a high impact). (**C**–**F**) Box plots demonstrating differences in the numbers of genetic variants between the analyzed groups of tumors for SNPs ((**C**) a high or moderate impact, (**D**) only a high impact) and non-SNPs ((**E**) a high or moderate impact, (**F**) only a high impact). Each box plot is additionally supplemented with the Kruskal–Wallis rank sum test (showing whether there is any statistically significant difference between the analyzed sets of variants) and the Wilcoxon rank sum test with continuity correction (the post hoc test applied to determine which tumor groups differed from each other). NS: not significant. Group sizes: BOT: *n* = 53; BOT.V600E: *n* = 23; lgOvCa: *n* = 10; hgOvCa: *n* = 139.

**Figure 2 ijms-25-10876-f002:**
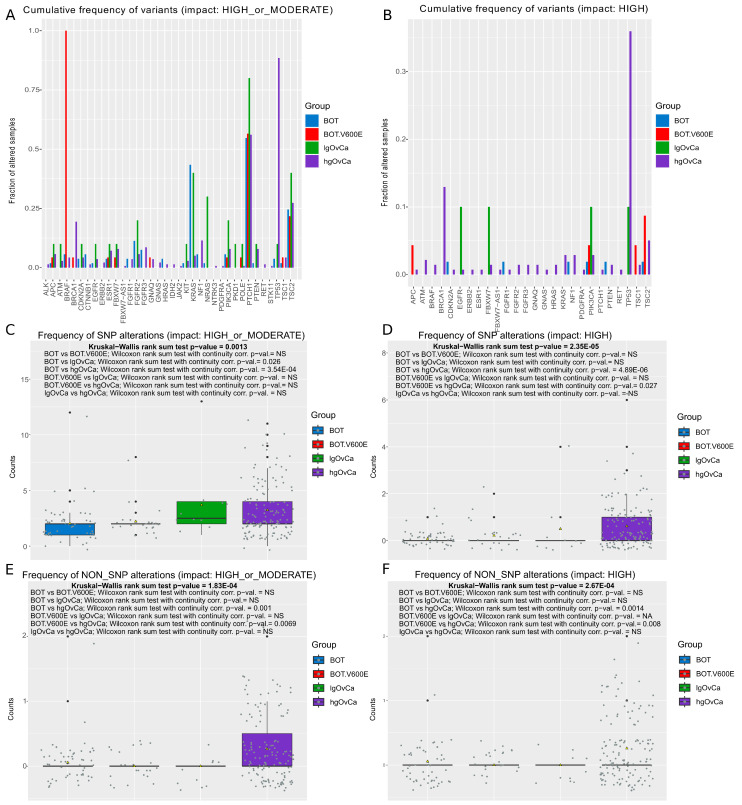
SNP and non-SNP variants—hot-spot gene panel. (**A**,**B**) Cumulative frequency of all variants (SNPs and non-SNPs combined) per gene in each tumor group ((**A**)—variants with a high or moderate impact, (**B**)—only the variants with a high impact). (**C**–**F**) Box plots demonstrating differences in the numbers of genetic variants between the analyzed groups of tumors for SNPs ((**C**) a high or moderate impact, (**D**) only a high impact) and non-SNPs ((**E**) a high or moderate impact, (**F**) only a high impact). Each box plot is additionally supplemented with the Kruskal–Wallis rank sum test (showing whether there is any statistically significant difference between the analyzed sets of variants) and the Wilcoxon rank sum test with continuity correction (the post hoc test applied to determine which tumor groups differed from each other). NS: not significant. Group sizes: BOT: *n* = 53; BOT.V600E: *n* = 23; lgOvCa: *n* = 10; hgOvCa: *n* = 139.

**Figure 3 ijms-25-10876-f003:**
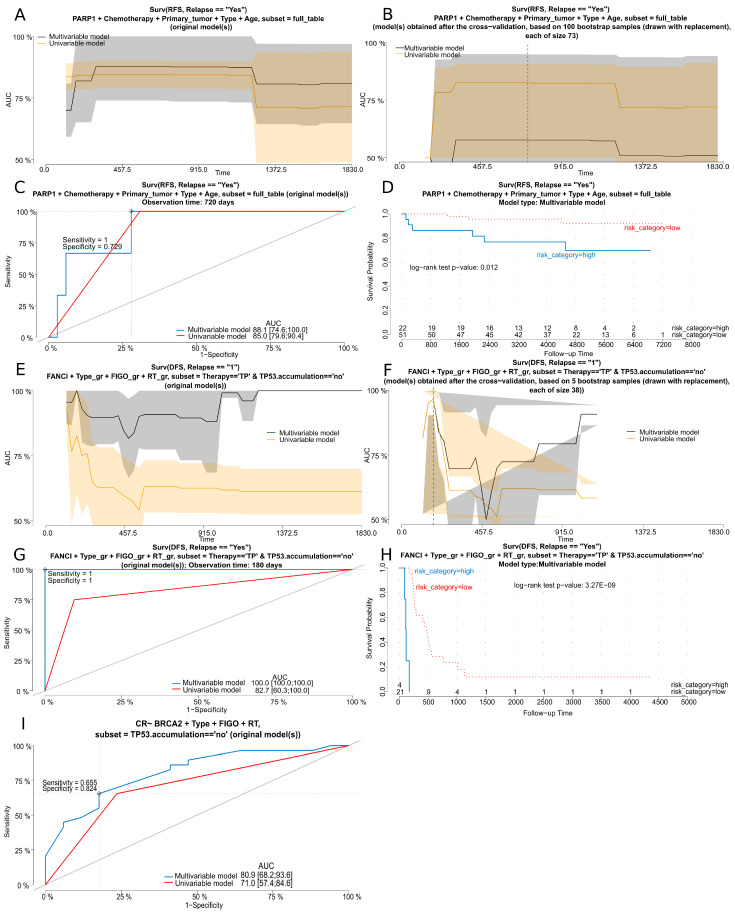
Cox and logistic regression analyses for selected genes. (**A**–**H**) Cox regression analysis results for the *PARP1* gene (RFS) in the whole BOTS group (**A**–**D**) and for the *FANCI* gene (DFS) in the subgroup of hgOvCa patients treated with the TP regimen and without *TP53* accumulation in their tumors (**E**–**H**). (**I**) Logistic regression analysis results for the *BRCA2* gene (CR) in the subgroup of hgOvCa patients without TP53 accumulation in their tumors. (**A**,**B**,**E**,**F**) AUC plots for uni- and multivariable Cox regression models obtained before (**A**,**E**) and after (**B**,**F**) a bootstrap-based cross-validation of the original dataset. The red dashed line indicates the same time point that was used to draw the time-dependent ROC curves (**C**,**G**). Optimal cutoff points for these ROC curves were calculated for the multivariable models based on the Youden index. Discrimination sensitivity and specificity values for cutoff points, determined for ROC curves in (**C**,**G**), are also provided. (**D**,**H**) Kaplan–Meier survival curves obtained for the patients divided into two categories (risk higher (high) or lower (low) than for the ROC curves’ (**C**,**G**) estimated cutoff point, based on the risk of relapse, calculated using the multivariable models. The Kaplan–Meier curves are supplemented with the results of the log-rank test as well. (**I**) ROC curves for uni- and multivariable logistic regression models. An optimal cutoff point for these ROC curves was calculated for the multivariable model based on the Youden index. Discrimination sensitivity and specificity values for this cutoff point are also provided. RFS—recurrence-free survival; DFS—disease-free survival; RT—residual tumor size; CR—complete remission; TP—taxane/platinum chemotherapy.

**Figure 4 ijms-25-10876-f004:**
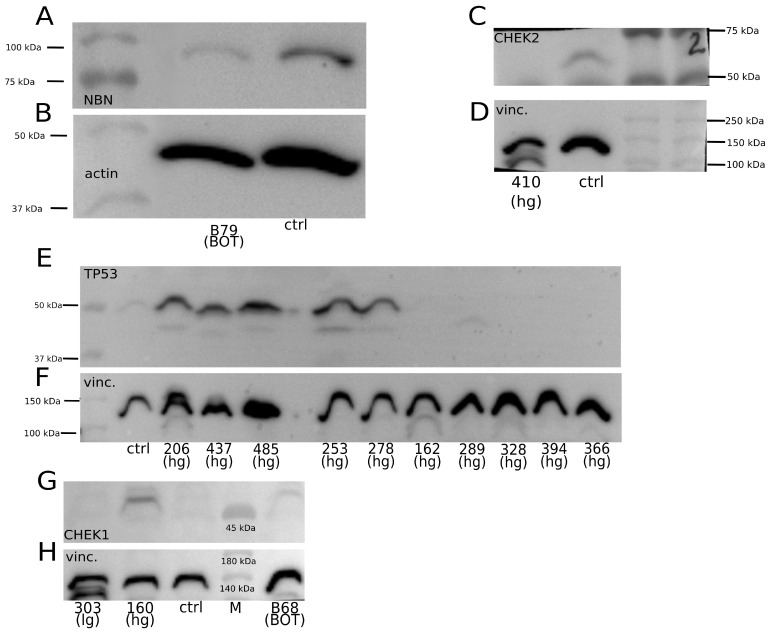
Selected genetic variants and their impact on the expression of corresponding proteins. (**A**) *NBN* chr8:g.89971217_89971221del (p.Lys219AsnfsTer16): 20% of reads with this sequence alteration (altered reads) in the B79 BOT sample. (**C**) *CHEK2* chr22:g.28695869del (p.Thr367MetfsTer15): 72% of altered reads in the 410 hgOvCa sample. (**E**) *TP53* missense variants: 206: chr17:g.7675085C>T (p.Cys176Tyr), 437: chr17:g.7673824C>G (p.Gly266Arg), 485: chr17:g.7676040C>G (p.Arg110Pro), 253: chr17:g.7673776G>A (p.Arg282Trp), 278: chr17:g.7673776G>A (p.Arg282Trp); TP53 non-SNPs with a HIGH impact: 162: chr17:g.7674900dup (p.Thr211AsnfsTer5), 289: chr17:g.7670686del (p.Arg342GlufsTer3), 328: chr17:g.7674241del (p.Cys242AlafsTer5), 394: chr17:g.7676078del (p.Pro98LeufsTer25), 366: chr17:g.7676041_7676042insTTTC (p.Arg110GlufsTer40). Altered reads: 206—64%; 437—72%; 485—71%; 253—84%; 278—63%; 162—67%; 289—52%; 328—43%; 394—40%; 366–50%. (**G**) *CHEK1* chr11:g.125625996G>A (p.Trp79Ter) in all three tumors. Altered reads: 303—18%; 160—69%; B68—49%. (**B**) Actin as a loading control, detected with a rabbit polyclonal anti-actin Ab, and (**D**,**F**,**H**) vinculin as a loading control, detected with a rabbit polyclonal anti-vinculin Ab. M—protein marker; vinc.—vinculin; ctrl—normal ovary; hg—hgOvCa; lg—lgOvCa; Ab—antibody.

**Figure 5 ijms-25-10876-f005:**
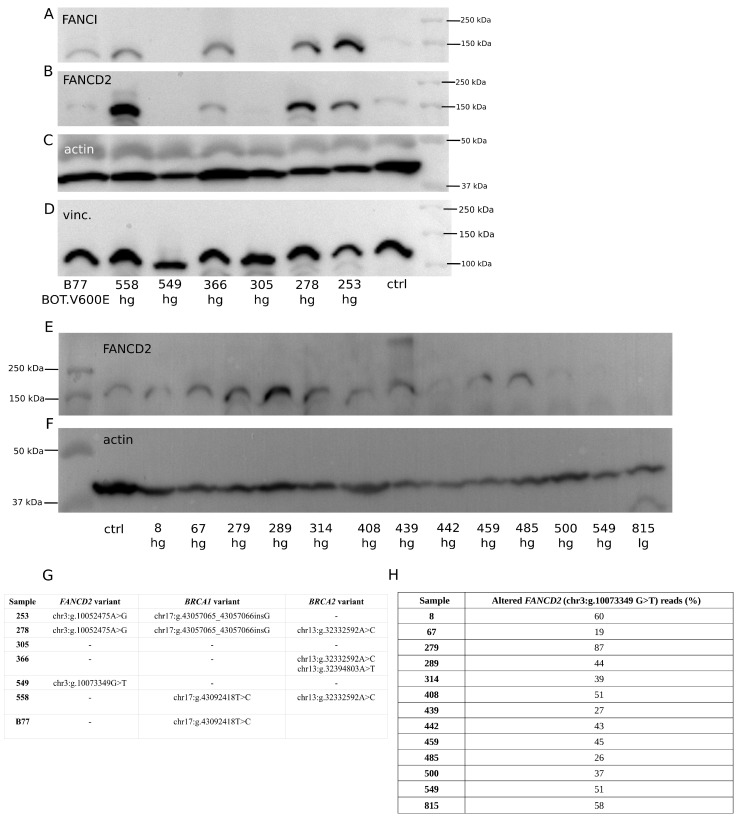
Frequent genetic variants in the *FANCI* and *FANCD2* genes and their impact on the expression of corresponding proteins. (**A**) the *FANCI* chr15:g.89285210C>T (p.Leu605Phe) variant; altered reads: B77—51%, 558—92%, 549—85%, 366—45%, 305—86%, 278—33%, 253—32%. No relationship between the percentage of altered reads and the protein level was observed. (**B**) FANCD2 expression for the same cases as in (**A**). (**E**) Expression of FANCD2 in the cases with the most frequently occurring *FANCD2* variant: chr3:g.10073349G>T (p.Gly901Val). No relationship between the presence of this variant, the percentage of altered reads (**H**), and the protein level was observed. (**G**) A table showing the occurrence of *FANCD2* and *BRCA1/2* variants in samples with the *FANCI* chr15:g.89285210C>T variant. (**C**,**D**,**F**) Loading controls. A rabbit polyclonal anti-actin or anti-vinculin primary antibody was used to detect actin (**C**,**F**) and vinculin (**D**), respectively. Ctrl—normal ovary; hg—hgOvCa; lg—lgOvCa.

**Table 1 ijms-25-10876-t001:** Genetic variants with a high or moderate impact significantly differentiating ovarian tumor groups, identified with two gene panels.

**44-GENE PANEL**						
**Impact HIGH or MODERATE**						
**Group Comparison and *p*-Value**						
**Gene**	**BOT vs. BOT.V600E**	**BOT vs. lgOvCa**	**BOT vs. hgOvCa**	**BOT.V600E vs. lgOvCa**	**BOT.V600E vs. hgOvCa**	**lgOvCa vs. hgOvCa**
** *TP53* **			5.67 × 10^−31^ (↑hgOvCa)		1.23 × 10^−18^ (↑hgOvCa)	1.8 × 10^−9^ (↑hgOvCa)
** *FANCB* **			9.71 × 10^−3^ (↑BOT)			
** *SEM1* **		2.51 × 10^−2^ (↑BOT)	1.01 × 10^−2^ (↑BOT)			
** *FANCA* **			2.61 × 10^−2^ (↑BOT)			
** *FANCD2* **			4.97 × 10^−2^ (↑hgOvCa)		1.52 × 10^−2^ (↑hgOvCa)	
** *BRCA2* **			1.47 × 10^−2^ (↑BOT)			
** *CHEK2* **			1.04 × 10^−2^ (↑BOT)			
** *MUTYH* **			1.44 × 10^−2^ (↑BOT)			
** *RAD50* **			2.83 × 10^−2^ (↑BOT)			
**Impact MODERATE**						
**Group Comparison and *p*-Value**						
**Gene**	**BOT vs. BOT.V600E**	**BOT vs. lgOvCa**	**BOT vs. hgOvCa**	**BOT.V600E vs. lgOvCa**	**BOT.V600E vs. hgOvCa**	**lgOvCa vs. hgOvCa**
** *TP53* **			3.48 × 10^−14^ (↑hgOvCa)		6.97 × 10^−9^ (↑hgOvCa)	1.64 × 10^−4^ (↑hgOvCa)
** *BRCA1* **			2.76 × 10^−2^ (↑BOT)			
** *FANCB* **			9.71 × 10^−3^ (↑BOT)			
** *SEM1* **		2.51 × 10^−2^ (↑BOT)	1.01 × 10^−2^ (↑BOT)			
** *MUTYH* **			3.8 × 10^−2^ (↑BOT)			
** *BRCA2* **			3.83 × 10^−3^ (↑BOT)			
** *CHEK2* **			5.94 × 10^−3^ (↑BOT)			
** *FANCA* **			2.61 × 10^−2^ (↑BOT)			
** *FANCD2* **			4.97 × 10^−2^ (↑hgOvCa)		1.52 × 10^−2^ (↑hgOvCa)	
** *RAD50* **			2.83 × 10^−2^ (↑BOT)			
** *PALB2* **			4.31 × 10^−2^ (↑BOT)			
** *ATM* **				3.62 × 10^−2^ (↑lgOvCa)		
**Impact HIGH**						
**Group Comparison and *p*-Value**						
**Gene**	**BOT vs. BOT.V600E**	**BOT vs. lgOvCa**	**BOT vs. hgOvCa**	**BOT.V600E vs. lgOvCa**	**BOT.V600E vs. hgOvCa**	**lgOvCa vs. hgOvCa**
** *TP53* **			1.25 × 10^−8^ (↑hgOvCa)		1.47 × 10^−4^ (↑hgOvCa)	3.08 × 10^−2^ (↑hgOvCa)
** *BRCA1* **			1.25 × 10^−7^ (↑hgOvCa)		6.01 × 10^−4^ (↑hgOvCa)	3.4 × 10^−2^ (↑hgOvCa)
**HOT-SPOT PANEL**						
**Impact HIGH or MODERATE**						
**Group Comparison and *p*-Value**						
**Gene**	**BOT vs. BOT.V600E**	**BOT vs. lgOvCa**	**BOT vs. hgOvCa**	**BOT.V600E vs. lgOvCa**	**BOT.V600E vs. hgOvCa**	**lgOvCa vs. hgOvCa**
** *TP53* **			1.01 × 10^−29^ (↑hgOvCa)		7.62 × 10^−18^ (↑hgOvCa)	2.35 × 10^−7^ (↑hgOvCa)
** *BRAF* **	1.52 × 10^−16^ (↑BOT.V600E)			1.08 × 10^−8^ (↑BOT.V600E)	1.08 × 10^−23^ (↑BOT.V600E)	
** *NRAS* **		1.1 × 10^−2^ (↑lgOvCa)		2.2 × 10^−2^ (↑lgOvCa)		2.22 × 10^−4^ (↑lgOvCa)
** *BRCA1* **			1.08 × 10^−4^ (↑hgOvCa)			
** *FBXW7* **			3.67 × 10^−2^ (↑hgOvCa)			
** *KRAS* **	6.44 × 10^−5^ (↑BOT)		2.58 × 10^−10^ (↑BOT)	5.13 × 10^−3^ (↑lgOvCa)		2.77 × 10^−3^ (↑lgOvCa)
**Impact MODERATE**						
**Group Comparison and *p*-Value**						
**Gene**	**BOT vs. BOT.V600E**	**BOT vs. lgOvCa**	**BOT vs. hgOvCa**	**BOT.V600E vs. lgOvCa**	**BOT.V600E vs. hgOvCa**	**lgOvCa vs. hgOvCa**
** *TP53* **			2.27 × 10^−15^ (↑hgOvCa)		1.66 × 10^−9^ (↑hgOvCa)	8.26 × 10^−5^ (↑hgOvCa)
** *BRAF* **	1.52 × 10^−16^ (↑BOT.V600E)			1.08 × 10^−8^ (↑BOT.V600E)	1.08 × 10^−24^ (↑BOT.V600E)	
** *NRAS* **		1.1 × 10^−2^ (↑lgOvCa)		2.2 × 10^−2^ (↑lgOvCa)		2.22 × 10^−4^ (↑lgOvCa)
** *KRAS* **	6.44 × 10^−5^ (↑BOT)		1.41 × 10^−11^ (↑BOT)	5.13 × 10^−3^ (↑lgOvCa)		1.11 × 10^−3^ (↑lgOvCa)
**Impact HIGH**						
**Group Comparison and *p*-Value**						
**Gene**	**BOT vs. BOT.V600E**	**BOT vs. lgOvCa**	**BOT vs. hgOvCa**	**BOT.V600E vs. lgOvCa**	**BOT.V600E vs. hgOvCa**	**lgOvCa vs. hgOvCa**
** *TP53* **			5.84 × 10^−9^ (↑hgOvCa)		1.35 × 10^−4^ (↑hgOvCa)	
** *BRCA1* **			3.98 × 10^−3^ (↑hgOvCa)			

*p*-values of the applicable (chi-squared or Fisher’s exact) test are included, followed by an arrow and the name of the group in which a given gene was more frequently altered (both written in brackets). In case of a lack of statistical significance, the corresponding cell is empty.

**Table 2 ijms-25-10876-t002:** The results of multivariable Cox and logistic regression analyses for the models with good discriminating capabilities (assessed based on their AUC values) that matched with corresponding univariable tests.

**44-Gene Panel**		
**BOTS**		
**RFS/relapse in the whole group of patients (full table)**	**HR [95% Cl]**	** *p* ** **-value**
** * PARP1 * **	6.82 [1.584–29.39]	0.01
**hgOvCa**		
**DFS/relapse in the subgroup of patients treated with TP and without *TP53* accumulation in tumors**	**HR [95% Cl]**	** *p* ** **-value**
** * FANCI * **	40.02 [3.784–423.133]	0.0022
Residual tumor <2 cm vs. no residual tumor (0 cm)	22.77 [2.061–251.608]	0.01
Residual tumor ≥ 2 cm vs. no residual tumor (0 cm)	34.1 [2.547–456.619]	0.0077
**CR in the subgroup of tumors without *TP53* accumulation**	**OR [95% Cl]**	** *p* ** **-value**
** * BRCA2 * **	7.06 [1.328–37.581]	0.022
**OS/death in the whole group of patients (full table)**	**HR [95% Cl]**	** *p* ** **-value**
** *BRCA2* **	0.58 [0.399–0.85]	0.005
Residual tumor <2 cm vs. no residual tumor (0 cm)	2.85 [1.654–4.903]	1.6 × 10^−4^
Residual tumor ≥2 cm vs. no residual tumor (0 cm)	3.75 [2.058–6.821]	1.55 × 10^−5^
**OS/death in patients with tumors without TP53 accumulation**	**HR [95% Cl]**	** *p* ** **-value**
** *BRCA2* **	0.42 [0.204–0.865]	0.019
Residual tumor ≥2 cm vs. no residual tumor (0 cm)	4.63 [1.348–15.883]	0.015
**OS/death in the subgroup of patients treated with TP**	**HR [95% Cl]**	** *p* ** **-value**
** *BRCA2* **	0.53 [0.337–0.84]	0.007
Residual tumor <2 cm vs. no residual tumor (0 cm)	2.97 [1.616–5.471]	4.6 × 10^−4^
Residual tumor ≥2 cm vs. no residual tumor (0 cm)	3.94 [1.944–7.986]	1.4 × 10^−4^
**CR in the subgroup of patients treated with TP and without TP53 accumulation in tumors**	**OR [95% Cl]**	** *p* ** **-value**
** *BRCA2* **	6.73 [1.047–43.239]	0.045
**PS in the subgroup of tumors without *TP53* accumulation**	**OR [95% Cl]**	** *p* ** **-value**
** *BRCA2* **	8.23 [1.509–44.836]	0.015
**PS in the subgroup of patients treated with TP and without TP53 accumulation in tumors**	**OR [95% Cl]**	** *p* ** **-value**
** *BRCA2* **	8.33 [1.251–55.476]	0.028
**OS/death in the subgroup of patients treated with TP and without TP53 accumulation in tumors**	**HR [95% Cl]**	** *p* ** **-value**
** *FANCF* **	0.15 [0.024–0.976]	0.047
Residual tumor <2 cm vs. no residual tumor (0 cm)	3.69 [1.159–11.74]	0.027
Residual tumor ≥2 cm vs. no residual tumor (0 cm)	7.75 [1.84–32.595]	0.005
**Hot-Spot Panel**		
**hgOvCa**		
**OS/death in the subgroup of patients treated with TP and with TP53 accumulation in tumors**	**HR [95% Cl]**	** *p* ** **-value**
** *TSC2* **	2.52 [1.191–5.329]	0.016
Residual tumor <2 cm vs. no residual tumor (0 cm)	2.86 [1.312–6.249]	0.008
Residual tumor ≥2 cm vs. no residual tumor (0 cm)	2.61 [1.104–6.146]	0.029

The best models, the discriminating capabilities of which are shown in Figure 3, are underlined. AUC values for each model are provided in a file named Appendix A. RFS—relapse-free survival; OS—overall survival; DFS—disease-free survival; TP—taxane/platinum chemotherapy; CR—complete remission; PS—platinum sensitivity; HR—hazard ratio; OR—odds ratio; CI—confidence interval.

## Data Availability

All data are available in the main text or Appendix A.

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
