# Peer review of "An Analysis of Genetic Polymorphisms in 76 Genes Related to the Development of Ovarian Tumors of Different Aggressiveness"

_ijms, 2024, doi:10.3390/ijms252010876_

Round 1

Reviewer 1 Report (New Reviewer)

Comments and Suggestions for Authors

The manuscript “The Analysis of Genetic Polymorphisms in 76 Genes Related to the Development of Ovarian Tumors of Different Aggressiveness” by Laura A. Szafron and co-authors to analyze genetic variants in crucial tumor suppressors and oncogenes in BOTS (with or without the BRAF V600E mutation), lgOvCa, and hgOvCa in two gene panels using next-generation sequencing. Then, they verified the existence of selected polymorphisms by Sanger sequencing. Finally, western blot analyses were carried out to check the impact of selected polymorphisms on the expression of the corresponding proteins. This study contributes to the molecular characteristic of ovarian neoplasms, demonstrating divergent polymorphic patterns pointing to distinct signaling pathways engaged in their development. Certain mutations seem to play an important role in BOTS without the BRAF V600E variant (KRAS) and in lgOvCa (KRAS and NRAS), but not in hgOvCa. Additionally, based on multivariable regression analyses, potential biomarkers in BOTS (PARP1) and hgOvCa (FANCI, BRCA2, TSC2, FANCF) were identified. Noteworthy, for some of the analyzed genes, such as FANCI, FANCD2, and FANCI, FANCF, TSC2 the status of BRCA1/2 and TP53, respectively, turned out to be crucial. These results shed new light on similarities and differences in polymorphic patterns between ovarian tumors of diverse aggressiveness. However, some concerns that must be taken into account before the work can be reconsidered for publication.

Comments

Can author confirm the gene polymorphisms of each gene? The restriction enzyme should be used. 

Comments on the Quality of English Language

Moderate editing of English language required.

Author Response

Dear Reviewer,

We greatly appreciate Your interest in this manuscript and would like to thank You for the suggestions and remarks that indeed increased the quality of the paper.

Ad 1. Can author confirm the gene polymorphisms of each gene? The restriction enzyme should be used.

Indeed, the results of the NGS analysis should be validated with other low-throughput methods to ensure the correctness of the entire analytical pipeline applied in the study and to prove that the bioinformatic methods used did not introduce any potential biases to the results. We are fully aware of the risks of drawing false conclusions if the high-throughput genetic analyses remain not validated with some other molecular methods. Therefore, we checked the quality of every DNA sample used in our study by measuring DNA amplification efficacy for two amplicons of different lengths by RT-qPCR. This in-house-developed method is described in detail in our previous paper [1]. If a sample did not pass this quality test, it was excluded from further analyses to diminish the risk of not detecting any polymorphisms due to the poor quality of DNA. Furthermore, it needs to be emphasized that in our bioinformatic workflow, all sequence variants less frequent than 10% were filtered out. This approach was to reduce the rate of false-positive hits caused by potential polymerase errors. Finally, we thoroughly analyzed the depth of NGS read coverage for every region in every gene and every sample assessed in the present study. If the mean coverage depth for a given region was lower than 5, this region was excluded from the analyses to diminish the risk of considering unevenly enriched DNA regions as non-mutated in samples with poor enrichment. Thanks to this multi-step filtering we managed to achieve high-quality NGS outcome, the reliability of which was later confirmed by PCR and Sanger sequencing for 8 selected novel genetic variants. Notably, in the meantime, one of these variants, in the TP53 gene, chr17:g.7670658_7670659insA, p.Lys351Ter was also confirmed by other researchers [2], thus providing the evidence for correctness and trustworthiness of our analytical pipeline. As to the confirmation of all genetic variants identified in our research, this would be a very tedious and money/time-consuming task. One has to remember the scale of our study, where 76 genes and approximately 400,000 nucleotides were evaluated in each specimen. In total, 2375 genetic variants of high or moderate impact were found, including 880 unique ones. One hundred and sixty-seven of them have not been previously identified by other researchers, involving 21 novel non-SNPs. It seems unfeasible to validate the presence of all these genetic variants with any low-throughput method, even if only the novel ones are to be verified. No matter what method was utilized for this validation, RFLP, AFLP, RT-qPCR or Sanger sequencing, it would be necessary to design hundreds or PCR primers to amplify the regions of interests. Considering that in our series of tumors there are 55 FFPE samples with a relatively low quality of DNA, this task seems even more challenging. As already proved by others [3], the use of restriction enzymes in the combination with FFPE material is tricky and laborious, demanding lots of optimizations and protocol modifications to obtain results of acceptable quality and reliability. In fact, in our previous scientific papers, before we managed to implement high-throughput sequencing techniques in our lab, we used to identify genetic polymorphisms with PCR-based approaches, such as SSCP [4–6]. Still, this workflow was much slower, much less sensitive and less precise than newly developed molecular methods, i.e., Sanger Sequencing, RT-qPCR, digital PCR and NGS, and therefore it was completely abandoned in our research. As for RFLP and other restriction-enzyme dependent techniques, their inferiority to Sanger sequencing and likely the other aforementioned methods, too, was reported to result from incomplete restriction enzyme digestion [7]. Given that genomic DNA, especially originating from old FFPE samples, is even more susceptible to cross-links and degradation that mitochondrial DNA [8], analyzed in the study cited above, the utilization of restriction enzymes to confirm genetic variants in our FFPE samples appears very limited. Moreover, some polymorphisms found in our study were located in low-complexity regions of the genome, rich in sequence repeats. In such cases, it would be extremely hard to find an appropriate restriction enzyme(s), offering such cleavage products that can be easily compared on either agarose or polyacrylamide gels.

Taking all the discussed questions into account, could You, please, reconsider the acceptance of our current article in its present form? In the future, we plan an in-depth verification of the clinical significance of some genetic variants found herein. In those upcoming studies, all selected polymorphisms will be precisely evaluated in every patient both qualitatively (by Sanger sequencing) and quantitatively (with either RT-qPCR or digital PCR).

Ad 2. Moderate editing of English language required.

The manuscript was, once again, thoroughly read and corrected by all the authors. Additionally, the entire text was scanned with the Grammarly app to eliminate all spelling mistakes, grammatical errors, and typos. Please, take a look at the text with tracked changes. Finally, if accepted, the article will undergo a language check performed by a professional English corrector employed at MDPI, which is a standard service, offered free of charge by this publisher.

References

1. Woroniecka, R.; Rymkiewicz, G.; Szafron, L.M.; Blachnio, K.; Szafron, L.A.; Bystydzienski, Z.; Pienkowska-Grela, B.; Borkowska, K.; Rygier, J.; Kotyl, A.; et al. Cryptic MYC Insertions in Burkitt Lymphoma: New Data and a Review of the Literature. PLoS ONE 2022, 17, e0263980, doi:10.1371/journal.pone.0263980.

2. Griffith, O.L.; Spies, N.C.; Anurag, M.; Griffith, M.; Luo, J.; Tu, D.; Yeo, B.; Kunisaki, J.; Miller, C.A.; Krysiak, K.; et al. The Prognostic Effects of Somatic Mutations in ER-Positive Breast Cancer. Nat. Commun. 2018, 9, 3476, doi:10.1038/s41467-018-05914-x.

3. Jovanovic, L.; Delahunt, B.; McIver, B.; Eberhardt, N.L.; Grebe, S.K.G. Optimising Restriction Enzyme Cleavage of DNA Derived from Archival Histopathological Samples: An Improved HUMARA Assay. Pathology (Phila.) 2003, 35, 70–74.

4. Moes-Sosnowska, J.; Rzepecka, I.K.; Chodzynska, J.; Dansonka-Mieszkowska, A.; Szafron, L.M.; Balabas, A.; Lotocka, R.; Sobiczewski, P.; Kupryjanczyk, J. Clinical Importance of FANCD2, BRIP1, BRCA1, BRCA2 and FANCF Expression in Ovarian Carcinomas. Cancer Biol. Ther. 2019, 20, 843–854, doi:10.1080/15384047.2019.1579955.

5. Dansonka-Mieszkowska, A.; Szafron, L.M.; Moes-Sosnowska, J.; Kulinczak, M.; Balcerak, A.; Konopka, B.; Kulesza, M.; Budzilowska, A.; Lukasik, M.; Piekarska, U.; et al. Clinical Importance of the EMSY Gene Expression and Polymorphisms in Ovarian Cancer. Oncotarget 2018, 9, 17735–17755, doi:10.18632/oncotarget.24878.

6. Rzepecka, I.K.; Szafron, L.; Stys, A.; Bujko, M.; Plisiecka-Halasa, J.; Madry, R.; Osuch, B.; Markowska, J.; Bidzinski, M.; Kupryjanczyk, J. High Frequency of Allelic Loss at the BRCA1 Locus in Ovarian Cancers: Clinicopathologic and Molecular Associations. Cancer Genet. 2012, 205, 94–100, doi:10.1016/j.cancergen.2011.12.005.

7. Tsai, N.-C.; Liou, C.-W.; Cheng, Y.-H.; Lien, H.-T.; Lin, T.-L.; Lin, T.-K.; Lan, M.-Y.; Hung, P.-L.; Wang, T.-J.; Lee, C.-H.; et al. The Establishment of a Molecular Diagnostic Platform for Mitochondrial Diseases: From Conventional to next-Generation Sequencing. Biomed. J. 2024, 100770, doi:10.1016/j.bj.2024.100770.

8. Merheb, M.; Matar, R.; Hodeify, R.; Siddiqui, S.S.; Vazhappilly, C.G.; Marton, J.; Azharuddin, S.; AL Zouabi, H. Mitochondrial DNA, a Powerful Tool to Decipher Ancient Human Civilization from Domestication to Music, and to Uncover Historical Murder Cases. Cells 2019, 8, 433, doi:10.3390/cells8050433.

Reviewer 2 Report (New Reviewer)

Comments and Suggestions for Authors

ijms-3158460-peer-review-v1

Laura A. Szafron et al.,

The Analysis of Genetic Polymorphisms in 76 Genes Related to the Development of Ovarian Tumors of Different Aggressiveness

Overview

The study is well-organized and manuscript is well written.

Comments

1. Were there any differences in results between the cases complicated with endometriosis/adenomyosis and those without them?

2. Were there any commonalities and/or differences in results between the lgOvCa and other malignancies such as endometrial cancer, colorectal cancer? Please discuss briefly (Quotation from past studies may be available).

3. Similarly, were there any commonalities and/or differences in results between the lgOvCa and Lynch syndrome?

4. In Tables, some genes were compared using odds ratio whereas others were compared using hazard ratio. Please explain the reason briefly.

Author Response

Dear Reviewer,

We greatly appreciate Your interest in this manuscript and would like to thank You for the suggestions and remarks that indeed increased the quality of the paper.

Ad 1. Were there any differences in results between the cases complicated with endometriosis/adenomyosis and those without them?

In the present study, the retrospective cohort of patients was used. For many of those patients, especially those collected in the 20th century, clinicopathological data were never digitalized, and they are available in a paper form only. The statuses of endometriosis/adenomyosis were not taken into account when the tumor samples were gathered for our study. Thus, it would be extremely difficult, and for some samples probably impossible, to assess this parameter after so many years. Still, we are really grateful for this remark and will try to assess whether ovarian tumors were accompanied by endometriosis/adenomyosis in our prospectively collected cohort of patients. Hopefully, in our future studies, we will be able to find some associations between these diseases.

Ad 2. Were there any commonalities and/or differences in results between the lgOvCa and other malignancies such as endometrial cancer, colorectal cancer? Please discuss briefly (Quotation from past studies may be available).

According to the reviewer’s suggestion, a short statement about the presence of KRAS, NRAS and BRAF mutations in colorectal and endometrial cancers was added in the Discussion section (see: lines 452-454 in the revised version of the manuscript).

Ad 3. Similarly, were there any commonalities and/or differences in results between the lgOvCa and Lynch syndrome?

Unfortunately, we are unable to tell whether genetic variants in the MLH1, MSH2, and MSH6 genes are associated with the development of lgOvCa, since none of these three genes, most strongly related to Lynch syndrome, was included in the gene panels analyzed in our study. Other genes linked to the risk of gynecologic cancer in Lynch syndrome, such as EPCAM and PMS2, were also absent in our gene panels.

Ad 4. In Tables, some genes were compared using odds ratio whereas others were compared using hazard ratio. Please explain the reason briefly.

The Cox and logistic regression models are different statistical approaches. The former is used to assess a time series, e.g. patient survival, that depends on one or many independent variables, e.g., the presence of genetic variants. The full name of the Cox model is ‘Cox proportional hazards model’, where the hazards are compared between the groups of samples with or without, e.g., the given genetic variant. The ratio between these risks is called hazard ratio (HR). By contrast, in logistic regression models, the dependent variable is always categorical (consisting of two categories), for example, complete remission present vs. absent. In this approach, odds, instead of hazards are compared, and their ratio is called odds ratio (OR).

This manuscript is a resubmission of an earlier submission. The following is a list of the peer review reports and author responses from that submission.

Round 1

Reviewer 1 Report

Comments and Suggestions for Authors

The present manuscript "The Analysis of Genetic Polymorphisms in 76 Genes Related to the Development of Ovarian Tumors of Different Aggressiveness" is an impressive and complex work a science team. The authors wanted to confirm certain selected gene variants through Sanger sequencing and to verify any possible correlation between the presence of polymorphism and the expression alterations of corresponding proteins. Studies have already shown variable expression levels of PARP1 in BOTS. PARP1 expression levels and activity could potentially serve as biomarkers to identify BOT patients who may benefit from PARP inhibitors. This is an area of active research, as the majority of studies have focused on high-grade ovarian cancers.

Some minor issues:

- pg 2, line 67-69: please rephrase for better clarity. Additionally, what do you mean by "BOTS (without BOT)"?

- line 82: how come is NGS suddenly mentioned. To this point abstract only mentions Sanger sequencing

- Related to previous issue: materials and methods should come before Results

Comments on the Quality of English Language

Please have the text read by a native English-speaker. Some syntax errors should need to be cleared.

Reviewer 2 Report

Comments and Suggestions for Authors

Based on my analysis of the article, there are some potential shortcomings of this study include:

1. The study included only ten low-grade ovarian carcinoma (lgOvCa) samples, which is a very small sample size compared to the other groups (53 borderline ovarian tumors without BRAF V600E mutation, 23 with the mutation, and 139 high-grade ovarian carcinomas). This small sample size for lgOvCa limits the statistical power and generalizability of comparisons involving this group.

2. The study used a retrospective set of non-consecutive ovarian tumor samples collected over 20 years (1995-2015). This non-consecutive sampling could introduce selection bias if the samples do not represent the overall patient population.

3. The study was conducted at a single institution in Poland, which may limit the generalizability of findings to other ethnic populations or geographic regions.

4. The study identified numerous genetic variants, but functional validation was only performed for a small subset. The clinical significance of many identified variants remains unclear.

5. The bioinformatic workflow filtered out variants with a frequency below 10% to reduce false positives. However, this may have resulted in missing some rare but potentially important genetic alterations.

6. The study analyzed bulk tumor samples, which may not fully capture the genetic heterogeneity within individual tumors.

7. As a retrospective study, it relies on previously collected samples and clinical data, which can introduce biases and limit the ability to control for confounding factors.

8. While the study mentions a minimum 3-year follow-up for surviving patients, more extended follow-up data could provide stronger insights into long-term outcomes and the clinical relevance of identified genetic markers.

9. The study does not mention using matched normal tissue samples, which could help distinguish somatic from germline variants more accurately.

10. The study focuses primarily on genetic alterations and does not explore epigenetic changes that may also play important roles in ovarian tumor development and progression.

These limitations should be considered when interpreting the results and planning future studies. Despite these shortcomings, the study provides valuable insights into the genetic landscape of ovarian tumors of different aggressiveness and identifies potential biomarkers for further investigation.

Comments on the Quality of English Language

minor
